
# Geological investigations of the Lizard District, Cornwall, England: 1818-1883

Carl N Drummond[1]

[1]Earth and Planetary Science, Department of Physics, Purdue University Fort Wayne, Fort Wayne, IN 46737

*Correspondence to*: Carl N. Drummond (drummond@pfw.edu)

**Abstract**. A suite of lower Paleozoic slab of oceanic lithosphere was obducted onto the southern margin of Avalonia during the Variscan orogeny is now exposed throughout the Lizard District of Cornwall, England. This complexly faulted and metamorphosed region of mafic and ultramafic rocks has been the subject of geological investigation for over two hundred years. Herein the most significant scientific contributions made over a sixty-five-year interval from 1818 to 1883 are reviewed. Early workers, including Ashurst Majendie, Adam Sedgwick, John Rodgers, and Henry De la Beche, conducted field-based studies of the region, making lithologic observations and mapping contacts between the major rock units. Subsequently, an intense phase of investigation into the processes and products of contact and regional metamorphism among primarily British geologists informed and inspired the field and microscopical studies of Thomas G. Bonney. Detailed consideration of the pioneering work of these 19th century geologists provides insights into their methodologies as well as their evolving understanding of the complex and enigmatic rocks of the Lizard.

## 1. Introduction

The Lizard district forms a headland of approximately 210 km$^2$ along the southern coast of Cornwall and is composed primarily of a suite of mafic and ultramafic rocks that have been subjected to complex structural deformation, metamorphic alteration, and subsequent mafic and felsic intrusion (Fig. 1). While tectonic and geochronologic uncertainties continue to be debated (e.g., Nutman et al., 2023; Mackay-Champion et al., 2024) the rocks of the region have, for the last half century, been understood to consist of segments of an obducted ophiolite suite thrust over the southern margin of Avalonia during the closing of the Rheic Ocean in the early stages of the Variscan orogeny. Such an interpretation was first suggested by G. A. Kirby (1975; 1978; 1979) in a series of papers of successively greater significance and impact. Modern interpretations of the Lizard indicate it consists of a sequence of oceanic crust and associated upper mantle rocks arranged such that the structurally deepest parts are exposed to the south while crustal lithologies are found to the north. In tectostraigraphic order (Fig.1), a series of sheeted dykes are found to occur at the top of the crustal Crousa gabbro which is in turn underlain by a highly sheared Moho transition zone, the Traboe cumulate complex which was unknown to early workers, all of which sit above a mantle sequence of serpentinized lherzolite, harzburgite, and dunite peridotites. Along the eastern margin of the peninsula the gabbros are cut by doleritic dykes while in the central and western parts of the peninsula the peridotites have been intruded by the Kennack gneissic granite. The entire ophiolite sequence is understood to have been thrust over a metamorphic sole

**History** of
Geo- and Space
**Sciences**
Discussions
consisting of the Landewednack amphibolite as well as a lower grade epidote amphibolite which has been interpreted to record the establishment of an inverted metamorphic gradient below the peridotite (Mackay-Champion et al., 2024). All of those rocks overly the highly deformed meta-sediments of the Old Lizard Head Series. The metamorphic sole is interpreted to have also been thrust over a structurally and lithologically complex mélange unit consisting of

fragments of the Man of War Gneiss, which is found in the tidal rocks along the shore of Lizard Point, the Mullion Island pillow lavas, as well as the Roseland breccias and spilites representing older oceanic crust which are now exposed beyond the northern margin of the complex (Fig. 1). Current tectonic interpretations are very much aligned with structural understandings of the classic ophiolite sequence of Oman (e.g., Cowan et al., 2014). Throughout the region the various lithologies are best observed along a series of high cliffs that encircle the peninsula, broken

occasionally by small sandy coves which provide access to many of the classic outcrops that have been the subject of study over the last two centuries.

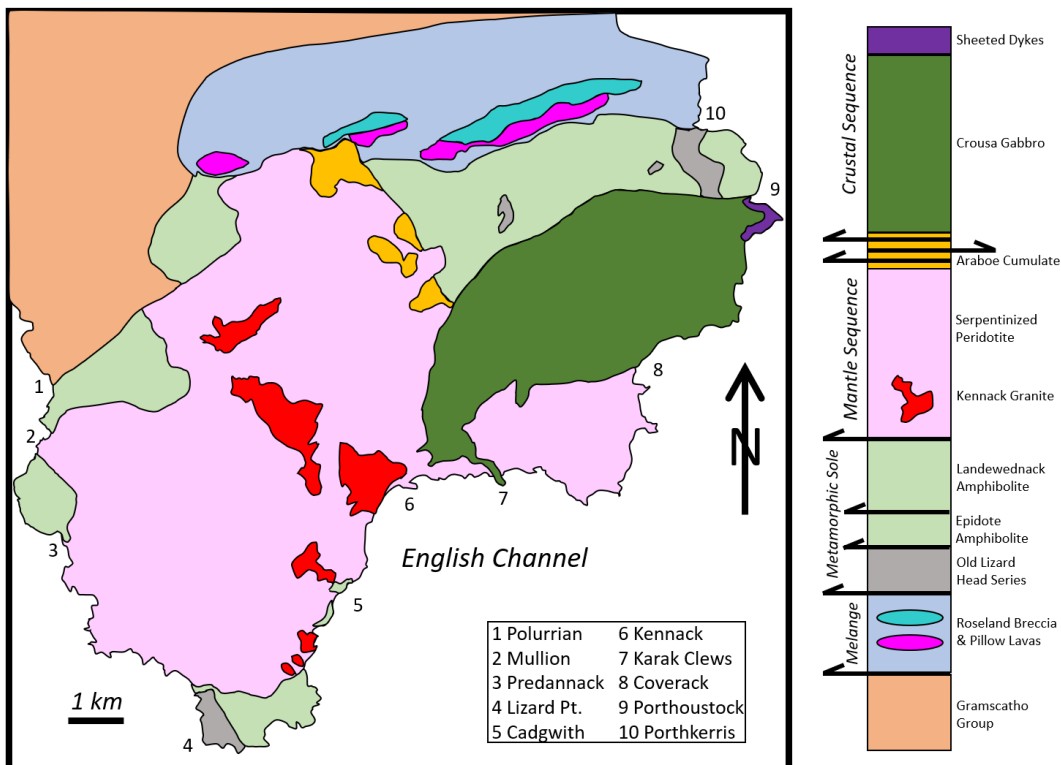

**Figure 1: Geological map and tectonostratigraphic section of rocks of the Lizard District. Various locations around the coast are identified as are the major tectonic segments of the ophiolite (crustal, mantle, metamorphic sole, mélange complex). Arrows in the column denote the presence of thrust and shear zones. Adapted from Macaky-Champion et al., 2024.**


Meaningful geological exploration of the rocks of the Lizard Complex began in the early 19[th] century. A comprehensive bibliography and accompanying brief synopsis of research up to 1940 was provided by John Smith Flett (1946) of the British Geological Survey wherein the contributions of early workers, as well as the somewhat



more extensive studies of later workers, were reviewed. A half century later at the annual conference of the Ussher
Society in January, 2000, Michael Styles of the British Geological Survey along with co-authors C. A. Cook and R.
E. Holdsworth of Durham University read a paper (Styles et al., 2000) which provided, in addition to a brief summary
of late 19th and early 20th century research, a review of those advances achieved since the establishment of a plate
tectonic framework for the interpretation of ophiolites. Interestingly, it was observed that several of those early
interpretations of age relationships among the rocks of the district had been proven to be correct by modern dating
methods, but often for reasons other than those proposed by the original workers (Styles et al., 2000). Recent
publications (e.g., Nutman et al., 2023; Mackay-Champion et al., 2024) only summarize of the history of research on
the rocks of the Lizard back to the landmark studies of David Headley Green (1964a; 1964b) wherein he interpreted
the Lizard District to have undergone regional metamorphism associated with the intrusion of an ultramafic body
(Styles et al., 2000). Given the relative brevity of previously published historical reviews, coupled with the general
omission of references to earlier studies within the recent geochemical and tectonic literature, it is interesting and
appropriate to review the early evolution of thinking regarding the geology of the Lizard Complex, particularly within
the context of the broader debate on the processes and products of metamorphism which occurred throughout the end
of the 19th and beginning of the 20th century (Drummond, 2026). The following study considers the first 65 years of
scientific study of the rocks of the Lizard, covering the interval from 1818 to 1883 while setting aside for future
consideration the intense controversies regarding the interpretation of these complex rocks that occurred at the end of
the 19th and beginning of the 20th century.

## 2. Earliest surveys

The earliest geological surveys of the Lizard Complex were conducted during the first decades of the 19th century,
prior to the introduction and widespread use of the polarizing petrographic microscope (Sorby, 1858; 1863; Hamilton,
1982). Workers were therefore limited to consideration of a series of field and hand-sample observations of the rocks
of the peninsula. It is useful to consider a general overview of the topographic and geographic context of the Lizard
district in order to inform a review of the region's geology. An elevated tableland plateau that forms inland portions
of the region contrasts sharply with outcrops found along the rugged cliffs of the coastline (Flett, 1946, pp. 1−11). As
such, early studies largely consisted of reports of observations conducted during perambulations along what is now
known as the Lizard Coastal Walk, with occasional forays down to sea level to inspect rocks exposed in the vicinity
of sheltered coves occurring along the shore. The most complete and influential early studies are reviewed in
chronological order to illustrate the progressively increasing understanding of the region's complex geology,
document how different workers approached the challenges presented by that complexity, and to summarize the status
of geologic knowledge at the dawn of the microscopic era of igneous and metamorphic petrology.

### 2.1 Majendie 1818

In 1818 Ashurst Majendie, a founding member of the Geological Society of London, conducted a study in which he
hoped "to offer to the Society an account of the boundaries and position of the serpentine formation, occurring in the





vicinity of the Lizard Promontory" (Majendie, 1818, p. 32). Following the coast in a counterclockwise direction,
Majendie worked from the Loe Pool on the west coast to the Helford estuary on the east, mapping the "junctions with
as much accuracy as the state of enclosure will permit" (Majendie, 1818, p. 32). He was able to fairly accurately
recognize the locations of the contacts between what are now known as the Old Lizard Head Series, the
Landewednneck amphibolite, the serpentinized peridotite of the Lizard Sequence, and the Crousa gabbro (Fig. 1).
Those observations allowed for the production of the first geologic map of the region. While detailed understandings
of the nature of the contacts, the relative ages of the bodies of rock, and their structural and petrogenic relationships
were not achieved, Majendie provided a sound set of field-based observations upon which future studies could build.
Given the uniqueness of the rocks, the structural complexity of the region, and the primitive analytical techniques
available, Majendie's map is remarkably congruent with modern surveys.

**2.2 Sedgwick 1822**

During the summer of 1819 Adam Sedgwick, recently appointed Woodwardian Professor of geology and Fellow of
Trinity College Cambridge, conducted a tour of Cornwall in the company of the Reverend W. R. Gilby also a Fellow
of Trinity and cousin of Dr. W. H. Gilby of Clifton who authored several studies of the geology of Bristol region
(Gilby, 1814; 1817). Sedgwick's "geological results of the tour were communicated to the Cambridge Philosophical
Society in two papers in 1820 and 1821" (Clark and Hughes, 1890, p. 213). The study was undertaken because "the
Lizard district differed so essentially from every other part of Cornwall" (Sedgwick, 1822, p. 291) and as such
Sedgwick hoped to establish the compositions of the rocks therein exposed and the nature of the contacts between
them such that his work would be "sufficiently complete to convey a correct notion respecting the character of the
great mineral masses which successively presented themselves" (Sedgwick, 1822, p. 291). Throughout his essay
Sedgwick used the somewhat poetic and alliterative term 'mineral masses' to denote the concept of the unique rocks
of the Lizard. In conducting their survey Sedgwick and Gilby followed the cliffs in a clockwise direction from the
Helford Estuary on the east to Mullion Cove on the west. They observed "the limits of these formations are therefore
traced with considerable accuracy in Mr. Majendie's map of the district … [however] I have as far as possible adopted
the orthography of the Ordnance Map. Still I have in more than one instance found great difficulty in describing the
locality of certain phenomena in such a way as to enable any one to observe them who may hereafter visit the coast"
(Sedgwick, 1822, pp. 293−294). This statement offered a clear description of the state of cartography as well as the
challenges of accurate geolocation along the rugged Lizard coastline in the early 19th century.

    Much as had Majendie (1818), Sedgwick was struck by how contacts between various lithologies were poorly
defined such that he observed that "those mineral masses which occur at the junction of two formations, or in parts of
the same formation where the mode of aggregation undergoes any sudden change, are often in an advanced stage of
decomposition" (Sedgwick, 1822, p. 297). Early workers were frequently unable to accurately ascertain the structural
relationships that existed along bounding contacts between major lithologic units and therefore their studies provided
very little useful structural interpretation of the Lizard beyond occasional dip−direction measurements. Among
Sedgwick's many field notes, while not directly related to the interpretation of the Paleozoic rocks of the Lizard, one
passage illustrated the acuity of his observations in that he and Gilby "twice crossed the high downs to the north and





north-west of Coverack" (Sedgwick, 1822, pp. 302−303), wherein they encountered "white quartz pebbles [that] abound in the alluvium which caps some portions of the downs" (Sedgwick 1822, p. 302). These largely unconsolidated sediments are now understood to be the Neogene Crousa gravels. Interestingly however, only a few other 19th century geologists (e.g., De la Beche, 1839; Budge, 1843) considered the origin of these younger sedimentary deposits perched upon the table-land, elevated some hundred feet above the level of the sea.

Moving southward, Sedgwick and Gilby "entered on the great formation of serpentine, which stretches from the east to the west coast, through a distance of six or eight miles, and occupies in superficial extent, about one-third of the peninsula … the local appearance of which has led to mistaken and contradictory opinions respecting the stratification of the whole formation" (Sedgwick, 1822, p. 303). This statement highlighted the difficulty in interpreting metamorphically altered igneous rocks exhibiting significantly variable and complex cumulate textures

as well as abundant rheomorphic fluxion structures. Working to the west of Black Head, Sedgwick and Gilby "expected that the rocks last described would have been succeeded by an extensive formation of greenstone, we were surprised to find them suddenly cut off by the re-appearance of serpentine" (Sedgwick, 1822, p. 305). The sojourners had encountered the promontory of Karak Clews, an eastward extension of a sheet of gabbro which "was wedged between two nearly perpendicular faces of serpentine" (Sedgwick, 1822, p. 205). This narrow ridge was subsequently

mapped inland for some distance and is now understood to be connected to the main body of the gabbro. Reaching Kennack Cove, Sedgwick "felt assured, from the experience of facts so many times repeated, that this complete degradation of the cliff must have been connected with some alternation among the masses which entered into this composition" (Sedgwick, 1822, p. 306). That is to say, Sedgwick had come to learn that where the cliffs became recessed and broken he could expect to observe a lithologic transition. Upon closer inspection of the coastal outcrops

to the west of the cove he lamented an "inability to convey, by verbal description, any thing like an adequate representation of the striking phenomena which rapidly succeeded each other" (Sedgwick, 1822, p. 306). The phenomena Sedgwick was trying to describe was the first occurrence of amphibolite along the eastern coast. An observation which was met with significant uncertainty in that "all these different mineral aggregates alternated in masses between which there seemed to be no fixed relation. They sometimes mutually penetrated each other; … [and

were] traversed by innumerable contemporaneous veins of green-stone; while, in the contiguous portion of the cliff, the green-stone so far predominated" (Sedgwick, 1822, p. 308). The generally two-dimensional aspect of the outcrops along the cliffs, the poor preservation of rocks in the vicinity of the fault-bounded contacts, and the absence of a robust theoretical framework within which petrogenic interpretations could be constrained collectively contributed to the complexity Sedgwick described as well as the lack of certainty contained within his interpretations.

Early 19th century geologic literature of the Lizard frequently commented upon the geomorphic feature known as the Devil's Frying Pan (Fig. 2). Located just below the village of Cadgwith it had been described by Sedgwick as "a magnificent natural amphitheater; formed by the decomposition of an irregular mass of serpentine, which had again intruded on the green-stone … above that portion of the cliff, where the sea has forced a passage into the chasm, a hard mass of serpentine stretches right athwart the gulf, and affords a natural bridge, by which we passed

over to the west" (Sedgwick, 1822, p. 310). The Cambridge geologists, likely in deference to their ordination, frequently omitted the satanic moniker for this location referring to it simply as the "Frying Pan". As evidenced by

modern digital maps of the British Geological Survey the geology of the Devil's Frying Pan is quite complex, with outcrops of the Lizard Peridotite, Landewednack Amphibolite, and Kennarck Granite all found within the circumference of the depression, the details of which were suggested but not fully understood by Sedgwick.

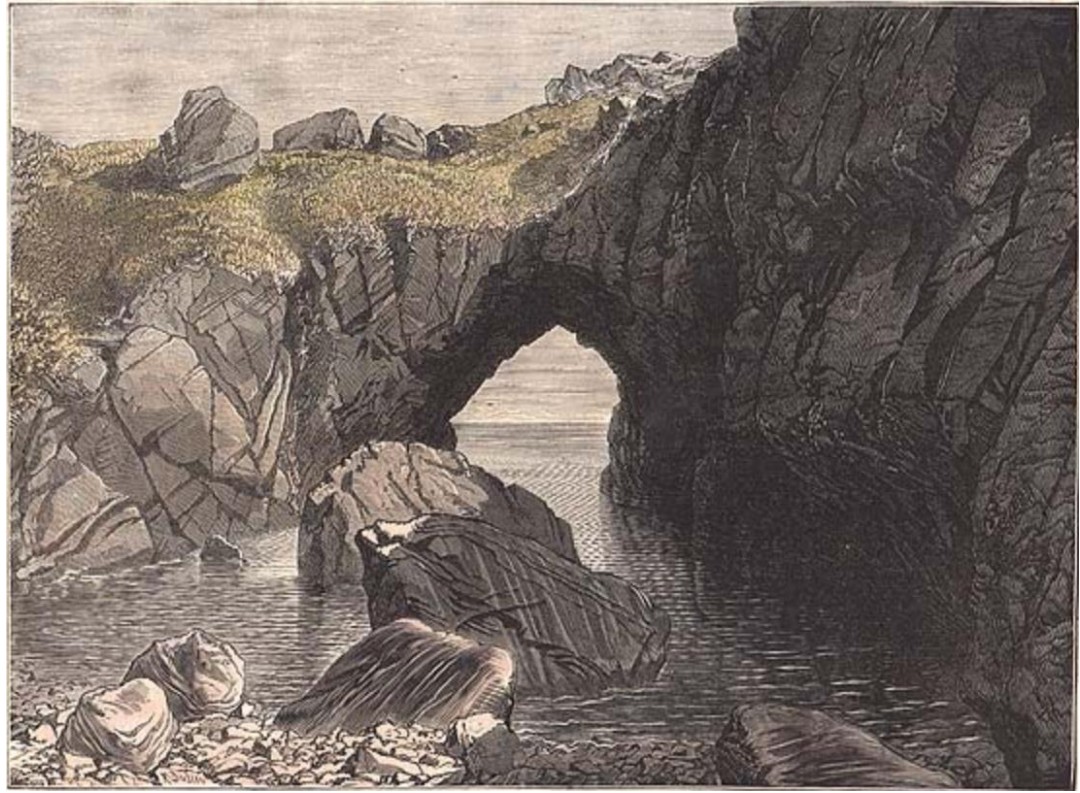


**Figure 2: Lithographic illustration of the Devil's Frying Pan, first published in the *Illustrated London News* in 1874 with later hand colouring. From the author's personal collection.**

The challenge to establish a set of satisfactory structural or stratigraphic relationships between the various units of the Lizard was sufficiently complex that Sedgwick along with Gilby "were led to conclude that the great
plateau of the Lizard was not composed of stratified rocks" (Sedgwick, 1822, p. 311). Rather, Sedgwick found relationships to be "too uncertain to be opposed to the clear evidence offered in the southeastern parts of the coast; where the alternating masses of greenstone and serpentine so often appear, like great wedges driven side by side into the escarpment without any order of arrangement whatsoever" (Sedgwick, 1822, p. 311). With this striking description of the complexity of the outcrop, Sedgwick captured the mystery, along with perhaps a bit of the awe, that the rocks
of the region inspired. These were, as he had noted, a set of rocks unlike any other in Cornwall.

Moving to the south, Sedgwick and Gilby encountered the rocks of Lizard Head, the southernmost point in England, wherein rather than clearly marking a transition from the serpentinized peridotite to the amphibolite their "examination of the coast was too much directed to general facts to allow us to enter into minute details" (Sedgwick, 1822, p. 313) of the complexities of the region. By only focusing on general field relationships, Sedgwick was able to



recognize that "beyond Balk Head commences the first formation of the green-stone slate in the whole district …
       [and] on both sides of Hot Point the formation continued to be characterized by a fine slaty texture" (Sedgwick, 1822,
       p. 314). Sedgwick's utilization of the term "green-stone" coupled with the adjective "slate" established a distinction
       between the amphibolite schists of the southeastern tip of the peninsula and the gabbro and sheeted dykes found further
       north which he had also referred to as "green-stone". This awkward use of terminology illustrated the challenges

inherent in interpreting the complex rocks of the Lizard without the aid of advanced petrographic techniques or the
       presence of a consistent classification of metamorphic rock types (e.g., Drummond, 2026). However, Sedgwick was
       able to recognize a lithologic transition between the eastern and western sides of the Lizard Head such that "the very
       imperfect evidence on which it had been attempted to establish the relation between the two formations, was not
       observed by us without some disappointment" (Sedgwick, 1822, p. 315). That is, while Sedgwick did not define

structural or stratigraphic relationships between the Old Lizard Head Series on the west and the Landewednack
       amphibolite on the east, he was able to recognize that there existed a clear distinction between those two bodies of
       rock. Due to the inaccessibility of the shore Sedgwick did not directly observe the outcrops of the Man of War gneiss
       that sat within the tidal range below the cliffs of the Lizard Head. Rather, he and Gilby reached an interpretation
       wherein those rocks exposed at sea level during low tide were thought to have "successively precipitated [fallen, not

crystallized] into the sea for want of an adequate support … [and] this we conceive to have been the mode of formation
       of the dangerous broken reef which stretches out so far to the south-west" (Sedgwick, 1822, p. 317). This
       misinterpretation of the Man of War gneiss as detritus rather than as an outcrop of an entirely different lithology is
       understandable given the challenges of observing those off-shore shoals on foot or by boat.

       Shifting his attention northward, Sedgwick noted "the western coast of the Lizard district abounds in grand

and picturesque features; yet in variety of composition, and in the interest which arises out of the junction of the rocks
       of different characters, it is greatly inferior to the eastern side of the peninsula" (Sedgwick, 1822, p. 319). Continuing
       his journey along the west coast, he provided descriptions of the rocks at Kynance Cove, Gue Graze, Pradanack Point,
       and Mullion Cove where "the island on the south-western side of the cove was evidently, at one time, a prolongation
       of the serpentine" (Sedgwick, 1822, p. 322). Having not visited Mullion Island, he did not recognize that the lithology

exposed was in fact significantly different than those occurring along the coast (e.g., Fox, 1895) and that the island
       consisted of a slice of older oceanic crust exhibiting spilite textures. Moving beyond the northern boundary of the
       Lizard, Sedgwick also recorded the presence of the Devonian schists of the Great Killas Formation in the
       neighborhood of Loe-Bar which are now known to be part of the Gramscatho group.

       Sedgwick's study concluded with a recapitulation of the lithologies encountered from the "heights above

Constantine, to the mouth of the Helford river, and from thence to Old Lizard Head, in the general direction of the
       coast … [then from] Old Lizard Head to Loe-Bar" (Sedgwick, 1822, pp. 326–327) and summarized his findings in
       the Lizard district such that "enough has been discovered to point that nature has at all times acted by general laws …
       [and] I have already expressed an option that the Lizard serpentine belongs to the later class" (Sedgwick, 1822, p.
       327) of transitional rather than primitive rocks within the lithologic classification system prevalent in the early 19[th]

century. That is, Sedgwick concluded that the serpentine was a transitional or secondary metamorphic rock rather than
       part of what was at that time termed the primary igneous rocks.

### 2.3 Rogers 1822

In the autumn of 1822, the Reverend John Rogers read a paper to the Royal Geological Society of Cornwall in which he noted that "the great serpentine district … extending from Porthall on the north-east coast to Mullion on the north-west, has been carefully examined by two intelligent Geologists" (Rogers, 1822, p. 416). The purpose of his study was to build upon those previous works in that "an accurate knowledge of a district of so complex a character can only be attained by repeated examination" (Rogers 1822, p. 416). Focusing his efforts on the eastern coast of the peninsula between Mande and Kildown Points, Rogers hoped to comment on "some particulars which have not been remarked either by Mr. Majendie or Professor Sedgwick" (Rogers, 1822, p. 416). Visiting several of the well-known localities along the coast, Rogers provided general observations of Porthkerris Cove, Manacle Rocks, Kennick Cove, and the Frying Pan. He concluded the paper by differentiating the rocks of the Lizard District into four groups: the Devonian clay slates found to the north, the amphibolite schist that at times he described as greenstone and at times as syenite, the gabbro that he frequently called diallage due to abundant pyroxene, and the serpentine which while noted in the paper's title but was only passingly remarked upon within the body of the text. Relative to the offerings of Majendie and Sedgwick, Rogers' paper was brief, and despite his stated intention of offering novel considerations, the small number of locations, most of which had already been more fully described in earlier reports, demonstrated a scarcity of new analysis provided by Rogers.

### 2.4 De la Breche 1839

In 1839 the first director of the Ordnance Geological Survey (later the Geological Survey of Great Britain) Henry T. De la Beche, in conjunction with a program of regional geological mapping, published an extensive report on the geology of southwest England (De la Beche, 1839). Running nearly 700 pages in length, his report offered a summary of the physiographic and geologic characteristics of the region as well as a catalog of associated agricultural, mining, and quarrying resources. The report also included copies of the royal charters granted to tin miners of Cornwall and Devin, first by King John on the 29th of October, 1201 and then by Edward the First on the 10th of April, 1305. Given the scale and scope of De la Beche's undertaking, only limited consideration was given to the geology of the Lizard district. However, of the several significant observations recorded, the first related to the contrast between various metamorphic rocks such that "in the Lizard district in Cornwall we again find mica slates … of very different character from those which constitute the great schistose systems of Corwall, Devon, and West Somerset. They occupy a very limited area near the Lizard Head, and have been noticed by the various writers on this part of Cornwal, among whom some diversity of opinion seems to prevail" (De la Beche, 1839, p. 29). Thus, De la Beche recognized the rocks of the Old Lizard Head Series and distinguished them from the Devonian schists found north of the peninsula. He also recognized that such a distinction had not been made by all previous workers. Further, he was able to recognize that "above, and indeed to a certain extent, interstratified with these talco-micaceous slates of the Lizard, we find well characterized hornblende slate … [which] supports the great mass of the Lizard serpentine, with an apparent passage of the one into the other in many places" (De la Beche, 1839, pp. 29−30). Noting many locations along the coast where the contact between these two bodies of rock can be found, De la Beche concluded "the hornblende slate and rock seem to have formed a basin into which the serpentine and diallage rock have been poured in a state of fusion" (De la



Beche, 1839, p. 30). This unique interpretation of the altered peridotite and gabbro as rocks of supposedly volcanic origin, while clearly incorrect, is perhaps understandable when considering the general synclinal geometry of rocks exposed along the western edge of the peninsula coupled with the difficulty in recognizing the complex faulting present throughout the region.

### 2.5 King and Rowney 1876

As noted by earlier compilations of scientific literature, contributions to the understanding of the Lizard district were relatively infrequent in the early to mid-nineteenth century. However, as subsequent decades passed, interest in the region's unique geology increased significantly. In 1876 professors William King and Thomas H. Rowney of Queens College Galway undertook an analysis of the serpentinite of the Lizard for the primary purpose of establishing the nature of its origin in that "Dr. Sterry Hunt maintains that serpentine is an original chemical precipitate" (King and Rowney, 1876, p. 281). The duo were not ones to shy away from a controversy and had engaged in a long-running and acrimonious debate with John William Dawson of McGill University on the organic versus mineralogical origin of what is now recognized as the pseudofossil *Eozoön canadense* (O'Brien, 1970; Adelman, 2007; Dolan, 2023) which had been correctly identified by the pair as interlayered calcite and serpentine of abiotic origin (e.g., King and Rowney, 1870a; 1870b). In addition to his extensive consideration of various occurrences of serpentine, King was also celebrated for his recognition that the Neanderthal was a species distinctly separate from modern Humans (King, 1864).

Returning to the Lizard, King and Rowney reached an interpretation that the "serpentine is in all cases the product of chemical changes or methylosis, effected in a preexisting mineral or rock of another kind, and analogous to pseudomorphism in crystalline solids" (King and Rowney, 1876, p. 251). By methylosis the authors meant those chemical changes associated with surficial weathering which occurred "near the surface of the earth, that is, from without" (Kinahan, 1884, p. 470). Evidence supporting the interpretation of this alteration was observed by them in the vicinity of Kynaance Cove where the "serpentine gradually changes into brown, pale yellow, and cream color … [and becomes] sinuously banded like chalcedony in agates" (King and Rowney, 1876, p. 383). In order to document this texture, as well as other petrographic features observed in rocks of the Lizard, King and Rowney created one of the most striking and unique illustrations to be found in 19[th] century geologic literature, a composite of 22 distinct images brought together into a single petrographic illustration which to the modern eye exhibits pronounced surrealistic qualities. By way of explanation of this complex composition, it was noted that "it will be understood that the figures represent things exhibited in a number of [different] specimens … [that] are always represented in their own matrix" (King and Rowney, 1876, p. 291). Further the "one most distinctive peculiarity which separates the Lizard serpentine from a number of other rocks of the kind … is the absence in it of the variety called ophite" (King and Rowney, 1876, p. 284) which was understood to be the intercalation of serpentine and calcite or dolomite often forming pseudo-organic structures such as *Eozoön canadense*. Their petrographic study of the serpentinite of the Lizard, so unlike those that proceeded it, or indeed any that followed, concluded with a review of the "structural simulations of organisms" observed to be present (King and Rowney, 1876, pp. 288−293) wherein a series of ovoid





and branching textures were reviewed and found to be of mineralogical rather than organic origin in what was clearly a recapitulation and extension of their earlier arguments against the interpretations of Davison.

**3. T. G. Bonney and the microscopical study of the Lizard**

The first full-scale study of the Lizard to significantly utilize modern petrographic techniques with the aid of a polarizing microscope was read to the Geological Society of London on May 23 1877 by the Reverend Thomas G. Bonney, Fellow of St. John's College, Cambridge (Bonney, 1877a). A companion paper presented the first geochemical study of the rocks of the region and was read by Wilfred Hudleston Hudleston (1877) at the same meeting.

Boney began his essay with a brief review of previously published literature on the Lizard District from Majandie to King and Rowney, noting that the later paper "appears to have been written with the view of calling attention to imitative organic, especially Foraminiferal structures. So far as my experience goes, any thing of this kind is very rare" (Bonney, 1877a, p. 884). Acknowledging the contributions of De la Beche and Sedgwick as "models of careful observation" (Bonney, 1877a, p. 884), he went on to suggest that the use of the "microscope has, I hope, enabled me

to explain several of those phenomena with which the appliances at their command could not deal" (Bonney, 1877a, p. 885). Generally following the routes of previous workers, Bonney "examined the coast … from Lizard Head to Mullion Cove on the west side, and from the same to Manacle Point on east, as carefully as circumstances admitted, and … also traversed the interior in two or three directions" (Bonney, 1877a, p. 885). Bonney acknowledged the challenges of conducting field work in the region such that "even at low water, progress at the base of the cliffs, where

possible, is often laborious and the state of the tide has to be carefully watched" (Bonney, 1877a, p. 885).

Five distinct lithologies were recognized by Bonney: hornblende schist, serpentinite, gabbro, granite, and greenstone. Here he used the term greenstone exclusively for those "dark augitic or hornblendic traps … restricted so far as I know to the east coast" (Bonney, 1877a, p. 885) which included the basaltic dykes found in the vicinity of Porthoustock as well as those exposed at Manacle Point. Bonney then provided a brief consideration of the hornblende

schist (Landewednack amphibolite) which he did not "profess to have minutely investigated as [he] was chiefly occupied with the rocks of presumed igneous origin" (Bonney, 1877a, p. 885). One ongoing peculiarity of Bonney's observations of the amphibolite was an erroneous interpretation of what he took to be relic sedimentary structures present in metamorphosed sediment such that the "the hornblendic bands alternate, occasionally exhibiting current-bedding … [and] foliation seems generally almost, or quite, parallel with the stratification" (Bonney, 1887a, p. 886).

Further, he disagreed with De la Beche's differentiation of the Old Lizard Head Series from the hornblende schist in that "as far, however, as I can make out, they only form a zone of some slight lithological peculiarities in the hornblende schist, into which they seem to pass almost insensibly" (Bonney, 1877a, p. 887). Much like previous workers, Bonney was unable to observe the unique lithology of the Man of War gneiss found off the coast of the Point of Polpeor.

**3.1 Survey of the west and east coasts**

One of the most important aspects of Bonney's report of field work along the coast of the Lizard was the efforts he made in trying to understand the nature of the lines of contact between the various lithologies. The first location



considered was Penthreath Beach where "a little chine runs along the line of junction. To make out the relations of the two rocks here is no easy task … after two or three visits and a most minute examination, I think I have succeeded …

the cliff on the north bank of the little chine is all serpentine; on the south the headland is all hornblende schist" (Bonney, 1877a, p. 887). Further northwest, in the vicinity of Kynance Cove, Bonney reported "the apparent interstratification of schist and serpentine on a smaller scale is due to the fact that a serpentinous mineral has been deposited by infiltration in the schist (as is commonly the case near a junction)" (Bonney, 1877a, p. 888), thus proposing the concept that the serpentine had, in a state of fusion, intruded the metasedimentary schist. Moving further

north and "passing Vellan Head, we continue to observe the apparently stratified structure in the serpentine, which sometimes even seems to mimic current bedding" (Bonney, 1877a, p. 890), thereby noting the presences of features which are now best interpreted as localized examples of fluxion structures. Moving up the west coast to "Ugethawr, on one side of George's Cove … I could not find the actual junction of the hornblende schist and serpentine on the rocky slope; but it is possible to scramble down to the water's edge, and there it can be discovered in a little sea-cave"

(Bonney, 1877a, p. 890) where he observed that "one or two thin tongues of serpentine are thrust into the schist within a foot or two of the junction. The serpentine is therefore intrusive" (Bonney, 1877a, pp. 890–891). Finally, having arrived at the northern limit of his field work along the west coast, Bonney noted "the celebrated Mullion cove is the end of a valley which very nearly defines the northern limit of this mass of serpentine; it is however, cut everywhere through the hornblende schist. The actual junction is masked by debris; so this also is inconclusive" (Bonney, 1877a,

p. 891). However, upon closer observation he was able to conclude "there can be no doubt that the serpentine is intrusive; on the northern side, against the cliff, is an included fragment of the schist" (Bonney, 1877a, p. 891). Documenting the complex and poorly exposed nature of the contacts that are now recognized as fault surfaces led to conflicting interpretations of the structural and temporal relationships between lithologies. Throughout the process of establishing these field observations, Bonney collected a suite of hand samples that he subsequently subjected to

detailed microscopic analysis.

Shifting to exposures along the east coast of the peninsula, in concurrence with Sedgwick's interpretation, Bonney noted "the sections on this coast are, on the whole, more complicated than those on the western. Commencing at the narrow cove of Pennanvose … the junction of the serpentine with the hornblende schist is well seen" (Bonney, 1877a, p. 892). Bonney again found the complexity of the contact between the amphibolite and the serpentine

perplexing such that in the vicinity of the Balk he observed variations in the degree of alteration of the peridotite, the least altered of which he took to be a gabbro, such that "the gabbro has penetrated again and again through the [hornblende] … so much that it is difficult to believe they come from a sedimentary rock" (Bonney, 1877a, p. 894). From his extensive study of the coastal sections below Landewednac, Bonney arrived at a set of four conclusions regarding the interpretation of temporal relationships among the lithologies observed (Table 1) and these

interpretations were supported by an unscaled field sketch of a portion of the exposure found at that location (Fig. 3) wherein a gabbro, in places foliated, was observed to intrude both the serpentine and the hornblende schist.


| 1 | The serpentine is an intrusive rock. |
|---|---|
| 2 | The hornblende schist was metamorphosed prior to serpentine intrusion. |
| 3 | The gabbro was intruded after serpentinization. |
| 4 | The black trap dyke was intruded last of all. |

**Table 1: Summary of Bonney's interpretations regarding the temporal relationships among the four lithologies observed in Fig. 3.**

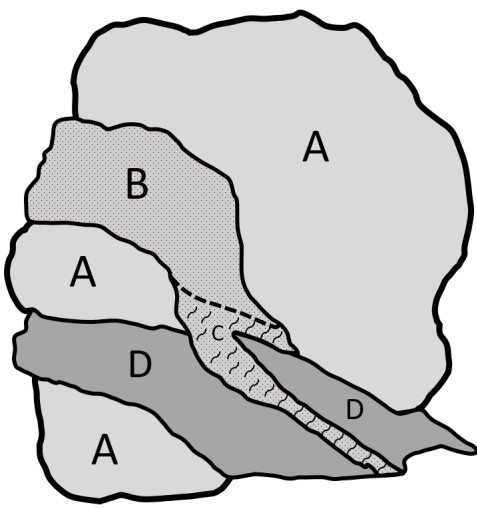


**Figure 3: Unscaled field sketch of lithologic relationships exhibited at the coastal section below Landewednac (A – serpentine, B – gabbro, C – foliated gabbro, D – hornblende schist). Adapted from Bonney (1877).**

Continuing north and east, Bonney advanced past Kennack Cove where "along the cliffs to the headland at Karak Clews … a bold headland formed by the extremity of a great elongated mass or broad dyke of coarse gabbro

… in which so many pieces of serpentine are entangled that it would often be easy to suppose the latter intrusive in the former" (Bonney 1877a, pp. 902–903). After making several more observations of the serpentine around Black Head and Chynhals Point, Bonney advanced to Coverack Cove where he observed that the serpentine was intruded by a network of dykes as well as two varieties of gabbro which he differentiated into an older and newer type. From these observations he was able to establish an "association of rocks in order of age" (Bonney, 1877a, p. 905) and much

as he had in the vicinity of Perranvose and the Balk, Bonney provided an unscaled field sketch to illustrate the age relationships between the older and newer gabbro as well as the trap rock (Fig. 4). This, as well as the example cited above, represented the first efforts to work out a temporal history of rocks of the Lizard based on detailed field observations. The survey of coastal outcrops concluded at Mancle Point and Bonney also provided brief summaries of two locations where at the first he observed the contact between the Devonian rocks and the Lizard serpentine. At

the location on Goomhilly Downs he observed several pits containing both banded and massive serpentine that was "extremely compact in texture, almost conchoidal in fracture, and very beautiful having a dull-reddish to greenish

groundmass with veins of bright red and yellowish steatite" (Bonney, 1877a, p. 914). It was from these small localized observations that coastal interpretations could be mapped into the peninsula.

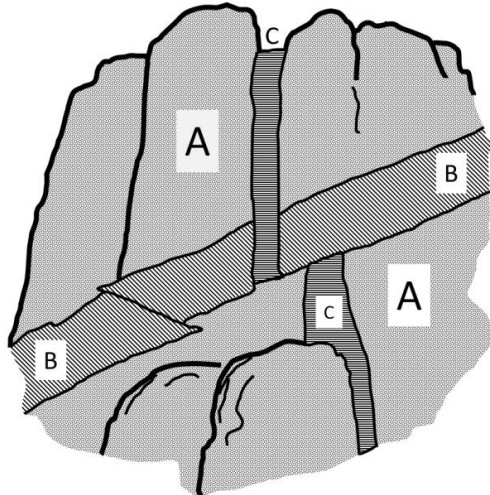

**Figure 4: Unscaled field sketch drawn from the base of the cliffs at Coverack Cove (A – older gabbro, B – newer gabbro, C – dark trap rock). Adapted from Bonney (1877).**

**3.3 Temporal relationships**

Having provided as complete and comprehensive a survey of the rocks of the peninsula as had up to that time been published, Bonney drew several conclusions which he offered as eight statements establishing a series of petrogenic

and temporal relationships among the identified lithologies (Table 2). First, Bonney concluded that the serpentine was originally an igneous rock that formed after the metamorphism of the amphibolite. Second, that the degree of serpentinization "is often rather illusory being due to the presence of an extremely small proportion of that mineral" (Bonney, 1877a, p. 915), and that the alteration of the peridotite to serpentine had been completed prior to later intrusions. In as much as Bonney had agreed with the interpretation offered by King and Rowney that the

serpentinization occurred through methylosis (that is near surface chemical reactions) the relationship of that process of alteration to subsequent igneous intrusions was not explained. Further, Bonney recognized three distinct types of intrusion to have occurred after serpentinization. To the west the serpentine was cut by granite while on the east a series of doleritic and basaltic dykes, along with two phases of gabbro (the older of which he understood to be the same age as the gabbro of Crousa Down) were thought to be younger than the serpentine. Finally, Bonney recognized

three forms of alteration within the gabbro: the conversion of feldspar to microcrystalline saussurite, the recrystallization of pryoxene (diallage) pseudomorphically to hornblende, and the serpentinization of olivine. While lacking a tectonic framework for understanding the relationships between the serpentine, gabbro, and the amphibolite, the temporal relationships worked out by Bonney generally align with modern interpretations.



| 1 | The serpentine was originally an intrusive rock. | | |
|---|---|---|---|
| 2 | That intrusion was posterior to the metamorphism of the hornblende schist. | | |
| 3 | To the west the serpentine has been broken through by granite dykes. | | |
| 4 | On the eastern coast it has been broken through by gabbros and trap dykes. | | |
| 5 | The gabbros were altered in three ways: | | |
| | *Feldspar → Saussurite* | *Diallage → Hornblende* | *Olivine → Serpentine* |
| 6 | The trap dykes were dolerites or basalts. | | |
| 7 | Serpentinization was complete before mafic intrusions. | | |
| 8 | The serpentinous aspect of the rock is often rather illusory. | | |

**Table 2: Summary of Bonney's eight interpretations regarding the petrographic and temporal relationships exhibited by rocks of the Lizard District.**

### 3.4 Microscopic examinations

Perhaps Bonney's greatest contribution was in the form of microscopical examinations which were undertaken "with the view of ascertaining, if possible, the original character of the rock which now constitutes the serpentine of the Lizard" (Bonney, 1877a, p. 915). All previous workers had limited their analysis to the mapping of the serpentine outcrops without significant analysis of the mode of formation of this distinctive mineral. By providing a review of his interest in the origin of the serpentine, Bonney noted that "a suspicion of the true nature crossed my mind in 1874, when examining a slide from the black serpentine near Cadgwith ... [while] a specimen collected in 1875 at Coverack Cove rendered the conjecture a certainty" (Bonney, 1877a, p. 916). In offering detailed petrographic descriptions of serpentine from fourteen different locations around the peninsula, Bonney (1877a, pp. 915–922) concluded "that the Lizard serpentine is an intrusive rock I have already shown ... the microscopic examination confirms the idea, which both *à priori* chemical considerations and the general aspects of the rock suggests, that it is an altered olivine rock" (Bonney, 1877a, p. 922). This alteration was well illustrated by the comparison of a sample from Coverack Cove with a specimen of lherzolite collected by Bonney from the Etang de Lherz, Ariège in the French Pyrenees (Bonney, 1877b) which showed that approximately two-thirds of the rock (Fig. 5) consisted of olivine broken up by a network of "serpentine veins run together sometimes like matted roots" (Bonney 1877a, p. 916). The process of serpentinization was attributed to the "gradual decomposition of the olivine by the action of slowly infiltrated water, during which the hydrous compound serpentine is formed" (Bonney 1877a, p. 922), an interpretation very much in line with the concept of methylosis put forward by King and Rowney which wa based in large part on the work of another Irish geologist George H. Kinahan (1884). However, such an interpretation it is not at aligned with modern understandings of serpentinization which are expressed by the moderate to high temperature metamorphic chemical reaction of olivine (forsterite) with water in the presence of excess silica to form the serpentine mineral lizardite.

$$3Mg_2SiO_4 + SiO_2 + 4H_2O \rightarrow 2Mg_3Si_2O_5(OH)_4$$

Forsterite   Silica   Water          Lizardite
In support of his conclusions as to the essential nature of olivine to the process of serpentinization Bonney highlighted the results of two recently published reports. Macpherson (1876) considered the alteration of olivine to serpentine in the Ronda Mountains of Andalusia, Spain, wherein Bonney found "some of his [Macpherson's] figures might have been taken from my Cornish slides, so great is the resemblance" (Bonney, 1877a, p. 923). Similarly, Samuel Allport in his comprehensive study of contact metamorphism in Cornwall had noted the occurrence in a sample

from Clicker Tor (east of the Lizard) as possessing "a variegated mass of pale green serpentine … [and] imbedded in this matrix there are numerous pseudomorphs after olivine" (Allport, 1876, p. 422). Based on his observations from across the Lizard, and supported by the work of others, Bonney became "rather suspicious of the common statements about the metamorphism of ordinary pyroxene and hornblendic rocks (i.e. those also containing a fair proportion of feldspar) into serpentine" (Bonney, 1877a, p. 932). Rather, in his view the presence of olivine was a necessity for the

occurrence of significant serpentinization. While several different reaction pathways for the formation of serpentine are now known, including reactions of enstatite to serpentine, the presence of abundant olivine is commonly considered to be an essential protolithic component.

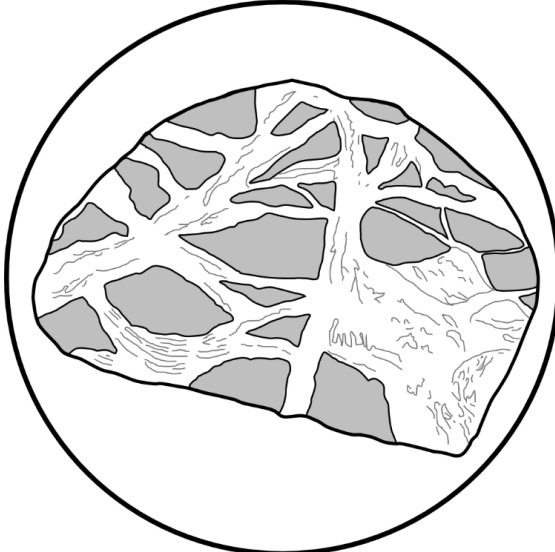

**Figure 5: Sketch of a portion of a microscopic slide of serpentine from Coverack cove. The granulated regions are**
**unchanged olivine while the light rights are fibrous serpentine. The original sketch was magnified 50 times. Adapted from Bonney (1877).**

### 3.5 Bonney's summary

Through the use of state-of-the-art petrographic examinations and extensive field-based observations, Bonney was able to arrive at several conclusions regarding the age relationships of the major rock groups of the Lizard. First, the

lherzolite peridotite had undergone serpentinization prior to the intrusion of the gabbro and granite. Second, that the "dykes of dark trap are the latest rocks of the peninsula" (Bonney, 1877a, p. 923). Contrary to the conclusions of De la Beche, Bonney was unable to establish relative age relationship between the gabbro and the granite, both of which he observed to have intruded the altered lherzolite in different parts of the peninsula. Finally, and again in opposition





to previous workers, Bonney concluded that the metamorphic rocks north of the Lizard were related to the
amphibolites and that both were derived from the metamorphism of sedimentary rocks of Devonian age.
Reconsideration of this interpretation became the primary motivation for the initiation of a subsequent study (Bonney
1883).

## 4. Bonney's survey of the Lizard schists

In the spring of 1882, accompanied by his friend the retired tutor at St. John's College, Cambridge, Reverend E. Hill
(e.g., Hill and Bonney, 1877; 1878; 1880), Thomas Bonney returned to the Lizard for the purpose of paying "special
attention to the structure of metamorphic rocks … as indicated on Sir. H. De la Beche's map from the Lizard Head to
Porthalla on the east; and to Polurrian Cove on the West" (Bonney, 1883, pp. 1−2). While his objective was to establish
age relationships "between the 'hornblende schists' and the slaty group" (Bonney, 1883, p. 2) that are found north of
the Lizard, he also set out to provide "some account of the two outlying masses of serpentine, which [he] had
previously been obliged to leave unvisited" (Bonney, 1883, p. 2). In concordance with his study of 1877, Bonney
established "that there is, in addition to the 'talco-micaceous' and 'hornblende schist' group already recognized, a
third or granulitic group" (Bonney, 1883, p. 2). Further he concluded that these three bodies formed a more-or-less
conformable sequence of metamorphosed sedimentary rocks with the talco-micaceous schists located along the
southwest coast of the Lizard Head being the oldest and the granulitic schists to the north and east the youngest. The
middle of the sequence was understood to be composed of the main body of the hornblende schist. East of Polpeor
Bonney identified a cove that "marks the junction of the micaceous with the hornblende series" (Bonney, 1887, p. 4)
where the boundary between the two rocks was interpreted to be a fault. Based on this and many other observations
Bonney concluded "faults abound in the Lizard district; they will be noted in almost every cove, inlet, or sea-chasm
(of which they are probably the cause). The generally uniform character of the rock makes it extremely difficult to
estimate the amount of vertical displacement; but as a rule, I believe it to be slight" (Bonney, 1883, p. 3), and he was
thus recapitulating and extending the observations Sedgwick had made six decades earlier. In interpreting the
structural and tectonostratigraphic relationship between the talco-micaceous schists of the Old Lizard Head Series and
the Landewednack amphibolite, and irrespective of his interpretation of a faulted boundary east of Polpeor, Bonney
maintained his "opinion previously formed that there is no real stratigraphic break between the two" (Bonney, 1887,
p. 4). This interpretation has not been substantiated by modern study of the Lizard such that a clear and significant
temporal and lithologic distinction is now made between the two lithologies of the Lizard Head.

### 4.1 Field and microscopic survey of the schists

The rocks of the Old Lizard Head Series are now interpreted to be a sequence of Cambrian sedimentary rocks
intercalated with felsic units of intrusive or volcanic origin (Mackay-Champion, et al. 2024) that comprise the lowest
portions of the metamorphic sole underlying the Lizard ophiolite (e.g., Jones, 1997). However, as noted above and
contrary to the interpretations of both Sedgwick (1822) and De la Beche (1839), Bonney was convinced they
represented the lowest part of a conformable sequence of metasedimentary rocks. Observing the talco-micaceous rocks





"at the south-west angle of the Lizard Head, called the Quadrant, we find the following series: – a thick mass of corrugated greenish schist, with 'cherty' bands, over which is a quartzose rock of rather gneissic aspect" (Bonney,

1883, p. 3). Along the west coast of the Lizard Head he noted "all or almost all these beds [are] in the lowest or 'talco-micaceous' group of De al Beche, and [I] think there is no important dislocation" (Bonney 1883, p. 3) between those rocks and the overlying hornblende schist.

East of the Quadrant, along the southern shore of the Lizard Head Bonney observed "a series of greenish micaceous schists of rather uniform character until we descend to the sea at Polpeor" (Bonney, 1883, p. 3). The rocks

observed at the base of the cliffs "consist of a green epidote schist in thick bands, alternating with brownish very micaceous schist, and with occasional lenticular hornblendic bands" (Bonney, 1833, p. 3). Bonney had passed across the faulted boundary between the Old Lizard Head Series and the overlying amphibolite and that at Polpeor he was observing the base of that series that consist of what is now interpreted to be an inverted metamorphic gradient within the sole (e.g., Mackay-Champion et al., 2024). Nearby Bonney found "a greenish schist with fairly well-marked

foliations, but very minute constituents, which becomes at times epidotic" (Bonney, 1883, p. 3). Moving further east to Bumble he noted that the outcrop "consists of hornblendic schist … and the same rock is exposed a short distance inland on the road from Lizard Town to Polpeor Cove" (Bonney, 1883, p. 4). From Bumble to Housel Bay, to Beast Point, and on to Hot Point, Bonney mapped the hornblende schist and noted that "it is impossible to resist the conclusion that, not withstanding the great amount of metamorphism we have a record of true 'current-bedding'

(whether by water or by wind) in the original constituents of the rock" (Bonney, 1883, p. 4). The small-scale foliations within the amphibolite that Bonney described as "here and there a kind of ripple drift" (Bonney, 1883, p. 4) were confusingly described both as false bedding and as relic primary sedimentary structures such that "lenticular bedding and even indications of current-bedding are far from rare, and the whole group gives one the impression of having been deposited by rather variable currents in waters of no very great depth" (Bonney, 1883, p. 5). This is an astonishing

interpretation of metamorphic fabrics as primary sedimentary structures which is at odds with modern understandings of the origin of amphibolite; a rock type associated with relatively high-grade metamorphism in which no evidence of primary structures from the mafic protolith are typically retained.

Bonney provided a field and microscopical description of a third series within the Lizard schists, which he termed the "granulite" group. While an explicit definition of this term was not presented, Bonney recognized two

distinct and often intercalated varieties in which one is a "rather finely granular pinkish grey rock, composed mainly of quartz and feldspar, with occasional specks of a dark mineral [hornblende] forming inconspicuous bands; the other a dark grey rock, in which the last named mineral predominates" (Bonney, 1883, p. 16). The granulitic schist was observed at Black Bay, in a quarry above Dolor Hugo, the Frying Pan, Ynys Head, and Caerleon Cove such that "along the eastern coast that the rather uniform and thick mass of hornblende schist passes upwards into a more

variable group, characterized by the presence of quartzo-feldspathic bands, which can hardly be less than two or three hundred feet thick" (Bonney, 1883, p. 7). Modern petrographic analysis suggests the variation Bonney observed in the upper portion of the hornblende schist can be understood in that "some samples are dominated by amphibole, whereas others are dominated by plagioclase. This variation likely reflects lithological heterogeneity within the mafic
protolith and/or differences in the degrees of partial melting" (Mackay-Champion et al., 2024, p. 9) associated with
the emplacement of the overlying peridotite. Such textures have been observed to depths up to 10 m below the contact.

**4.2 Other observations and interpretations**

Bonney concluded his essay with three relatively brief sections, two of which evaluated the age and composition of
the amphibolite protolith while the third provided details of two occurrences of serpentine not previously considered.
Bonney's 1883 study served to reaffirm an interpretation he had set forward in his 1877 paper that the rocks now
known as the Old Lizard Head series and the Landewednack Amphibolite recorded the metamorphism of a more-or-
less conformable sequence of sedimentary rocks. However, Bonney did acknowledge that J. Beete Jutes, Director of
the Geological Survey of Ireland and Lecturer at the Royal College of Science in Dublin, had posited that "some of
the hornblende schists … having been basaltic tuffs" (Bonney, 1883, p. 19) prior to their metamorphism. Further,
Bonney noted that Huddleston's chemical analysis of 1877 supported Jukes' interpretation of a mafic protolith.
However, "while examining the hornblende groups I had always present to my mind the possibility that the more
massive varieties might be really metamorphic igneous rocks" (Bonney, 1883, p. 19) ultimately, he rejected such a
possibility. While Bonney's interpretations of the conformity and protolithic composition of the metamorphic series
of the Lizard has been shown by subsequent workers to be incorrect, one of the primary motivations for the 1883 study
of the metamorphic series was to ascertain if the metamorphic rocks of the Lizard were of similar age to those of the
Devonian metamorphic rocks that bound the district to the north. In order to offer a revised interpretation, Bonney
reviewed several hypotheses of the age of metamorphic protoliths from other regions of England and Scotland.
Rejecting his prior concepts regarding the age of the metamorphic rocks of the Lizard, he summarized his revised
views in that "I consider that we are not yet in a position to offer a definite classification of the members of the
Archæan series, I will say no more than that I consider these Lizard rocks to belong to some part, and that by no means
the latest, of the record of that immense period of time" (Bonney, 1883, p. 21). This interpretation of a major lithologic
and temporal distinction has been substantiated by subsequent workers, although the amphibolite rocks of the Lizard
are now understood to be of early Paleozoic rather than Archean age.

Bonney's second study of the Lizard District concluded with a review of previously unvisited exposures of
the serpentine in the northeast regions of Polkerris and Porthalla. At those locations he found additional confirming
evidence that the serpentine had originally been an olivine-rich peridotite with locally abundant enstatie and augite.
Noting that "in the course of my work, I reexamined almost all the junctions described in my former paper, and have
no hesitation in affirming that there is abundant proof of the intrusive character of the serpentine in the Lizard District"
(Bonney, 1883, p. 22). A corollary of his interpretation of the Lizard serpentine as an intrusive igneous rock was his
concurrence with the conclusions of King and Rowney (1876) that Sterry Hunt's interpretation of serpentines as
having formed in ways similar to sedimentary bodies was incorrect. With a touch of British pride, he noted that he
was continent in the knowledge "that Dr. Sterry Hunt has never seen the Lizard District" (Bonney, 1883, p. 23).

## 5. Conclusions

The first significant geological surveys of the Lizard District were undertaken in the early 19th century. During the from 1818 to 1883 the methods and sophistication of scientific studies underwent significant evolution. The earliest surveys set out to map the positions of the major lithologic units along the coast of the peninsula. Adam Sedgwick, building on the work of Ashurst Majendie, conducted the first comprehensive survey. As would be the case with later workers, Sedgwick was confounded by the complex and frequently highly fractured and weathered contacts between
the major units and the establishment of firm tectonostratigraphic and structural relationships remained illusory. Recognizing the gabbro, serpentine, and hornblende schist as distinct units, unique to the Lizard peninsula, Sedgwick also concluded that the schists of the Old Lizard Head series were neither related to the Devonian schists that comprised much of central Cornwall nor to the adjacent but compositionally distinct hornblende schists.

Following the work of Sedgwick, John Rodgers focused his study primarily on the east coast of the peninsula
where he revisited many of the locations described by earlier workers. Almost two decades later, as part of a larger survey of Cornwall, Henry De la Breche in addition to differentiating the Old Lizard Head series from the hornblende schist, posited that the serpentine and diallage (gabbro) flowed into a basin formed by the older rocks. This interpretation required that the rocks had been formed in place with little or no deformation after lithification and was at odds with abundant evidence for fault bounded contacts between the major units of the district. Three decades later
the Irish geologists William King and Thomas Rowney visited the Lizard for the purpose of studying micro-textures of the serpentine. Their work can be best understood as an effort to object to Sterry Hunt's interpretation of serpentine as having a sedimentary origin as well as an effort to support a position they had taken in opposition to John Dawson's biological interpretation of the pseudo fossil *Eozoön canadense*. While providing an interesting chapter in that great debate, King and Rowney did little to significantly advance understandings of the geology of the Lizard District.

The first worker to combine detailed field observations and mapping with petrographic analysis utilizing the polarizing microscope was Thomas Bonney. Building on previous studies, Bonney set out to establish temporal relationships among the major units of the Lizard utilizing cross cutting relationships and the law of included fragments. Much as it had for Sedgwick and others, the ubiquity of fault surfaces in the most important outcrop localities led to uncertainties in interpretation. Oddities of Bonney's first study, which were perpetuated in his
subsequent paper, included his interpretation of the hornblende schists as containing relic primary sedimentary structures as well as his conclusion that the Old Lizard Head series was the stratigraphically lowest part of a metamorphosed conformable sedimentary sequence of the hornblende schist. With regard to the serpentine, Bonney became convinced based on detailed petrographic observations that the presence of olivine, rather than enstatite, was necessary for processes of serpentinization. Further he agreed with King and Rowney that the hydration reaction had
occurred through low temperature methylosis. His interpretation of the schists of the Lizard as being of similar age as the Devonian schists that bound the district to the north became a primary motivation for his subsequent study of 1883. In that later paper, by focusing on the metamorphic, rather than the igneous rocks of the Lizard, Bonney was able to refine his understanding of age relationships which led him to conclude the hornblende was much older than the Devonian schists. Despite this advance, he continued to fail to recognize the tectonostratigraphic relationship between
the Old Lizard Head series and the hornblende schists and he reinforced previous misidentification of schisosity and





small scale folding within the hornblende as primary sedimentary structures. Despite these drawbacks, Bonney made major advances in the understanding of the petrology and temporal history of the rocks of the Lizard through detailed field and microscopical observations.

With the rapid increase in the analysis of regional and contact metamorphism by a large number of geologists throughout England, Ireland, and Scotland (e.g., Drummond, 2026), the stage was set by these early workers for an expansion of study of the complex and unique rocks of the Lizard which was marked by vigorous debate during the late 19th and earliest 20th centuries regarding the igneous and metamorphic processes that acted upon them.

**Competing interests**. The author has declared that there are no competing interests.

**Acknowledgements**. The indefatigable support of the librarians and staff of the Walter E. Helmke library in supplying materials through document delivery and interlibrary loan is gratefully acknowledged. Without their efforts this work could not have advanced.

**Reviewer statement**.

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
