# Peer review of "Geological investigations of the Lizard District, Cornwall, England: 1818–1883"

_History of Geo- and Space Sciences, 2025_

## Referee Comment (RC1)

REVIEWER COMMENTS   Reviewed by:  William R. Brice (wbrice@pitt.edu).  27 August 2025

Article Reviewed:  **Geological investigations of the Lizard District, Cornwall, England: 1818-1883** by Carl N Drummond  https://doi.org/10.5194/hgss-2025-5.

GENERAL:  Very nice analysis and well organized.

SUGGESTIONS AND COMMENTS:

1.  When quoting from another publication it is my understanding (based on the *Chicago Manual of Style*, 13[th] edition, sections 10.36 to 10.49) ellipses are used to indicate omitted material.  I think it would make the quotes in the text clearer if ellipses were used, *e.g.* pages 3 and 4 with the quote from Majendie – ellipses added in red:

> In 1818 Ashurst Majendie, a founding member of the Geological Society of London, conducted a study in which he hoped "…to offer to the Society an account of the boundaries and position of the serpentine formation, occurring in the vicinity of the Lizard Promontory…"  (Majendie, 1818, p. 32).

Also, shouldn't the *et al.* when used in a citation be in italics?  Same for *e.g.*?  That is not the case in the paper.

SPECIFIC QUESTIONS/SUGGESTIONS:

1.  Page 9, line 262 – mention of the work of T. Sterry Hunt (the first of several), but no reference is given.  It would be nice for the readers to have a reference or two to Dr. Hunt's work related to the subject under discussion.

2.  Page 9, line 279 – reference to a figure in the work of King and Rowney 1876 – would it be possible to add that figure to this article so the readers can see the "..surrealistic qualities."?

3.  Page 11, line 328 – part of a quotation with ( ) in it.  I assume the ( ) is in the original, but it would be clearer to the reader if it was stated in the reference that the ( ) is in the original, and not added by the author of the article.  Just keeps things clearer.

4.  Page 12, caption for Figure 3 – is the citation 1877a or 1877b?  There are two Bonney 1877 references from which to choose.

5.  Page 13 – caption for Figure 4 – same as number 4 above; is the citation for 1877a or 1877b?

6.  Page 15 – caption for Figure 5 – same as numbers 4 and 5 above; is the citation for 1877a or 1877b?

7.  Page 16, lines 466 and 474 – citations for "Bonney 1887."  There is no "Bonney 1887" listed in the references.  If this is not a typo and should be Bonney 1883, then the reference for Bonney 1887 should be added to the reference list.

8.  Page 17, line 491 – citation for "Bonney 1833."  There is no Bonney 1833 listed in the reference list.  If this is not a typo and should be Bonney 1883, then the reference for Bonney 1833 should be listed in the reference list.

9.  Page 18, line 545 – possible typo; "…abundant enstatie and augite."  Should that mineral be enstatite?

10.  Page 19, lines 555 and 556 – "…undertaken in the early 19th 555 century.  During the from 1818 to 1883 the methods and sophistication…"  Seems like a word is missing from the sentence beginning with "During."  Perhaps it would read more smoothly like this:  "During the time from 1818 to 1883…" or "During the period from 1818 to 1883…"

11.  Page 20, line 599 – where is the Walter E. Helmke Library located?  Is it at Purdue or in England?

---

## Author Response (AR1)

Summary of responses to reviewer comments. HGSS 2025-5

I am very grateful for the clear, direct, and highly actionable comments and suggestions of Professor Brice and the anonymous reviewer. Their input has, I believe, significantly enhanced the paper. In order to provide a complete summary of the changes and modifications I have undertaken as a result of their input I provided the following summaries.

Brice review:

Suggestions and Comments –

I have not adopted the first suggestion offered. I think embedding ellipses at the beginning and end of each quotation as suggested in the example will 1) break up the rhythm and readability of the paper and 2) add unnecessary length. However, of course if the Editor requests, I will follow this suggestion.

Secondly, in my review of recent papers published in HGSS, I noticed that house style is not to use italics for e.g. and et al. If I am mistaken, I will change them to italics if I am incorrect.

Specific Questions/Suggestions –

1) T. Sterry Hunt citation, I have added a reference to Logan et al. 1863 to page 9 which is the example cited in King and Rowney 1876.
2) Both reviewers requested that I include the plate from King and Rowney and I have done so on page 10.
3) I have removed the ( ) from the quotation. If the Editor feels they should be included, I can easily reinsert.
4, 5, 6 etc.) these all should have been 1877a, corrections have been made
7)  This is a typo, corrected to 1883
8)  Spelling of enstatite corrected
9)  the words "the interval" inserted
10) Helmke Library is on my campus in Fort Wayne, I have made that notation.

Anonymous Review:

Two Primary Comments –

1) I have addressed the context and significance of Bonney's work more clearly in the abstract, introduction, and in the early parts of section 3 on page 11. Bonney's work serves as a bit of a pivot between the field-based observations of the early 19th century and the intensive debates about the Lizard that ensued in the decades of the late 19th and early 20th century. He is, so to speak, a bridge between those two phases of work. I had initially considered including a comprehensive summary of post-Bonney work but found that the manuscript – which is already quite long – would have likely doubled in size. As such I chose to have the work of Bonney be the "climax" of this paper and then something of the "jumping off point" for a future contribution. I hope I have addressed the reviewer's concern sufficiently.

2) This is a challenging comment to address for several critical reasons. The reviewer asks "what did others make of ophiolites and metamorphic rocks at this time?" First, the concept of an ophiolite sequence was unknown prior to the 1960s. Likewise, the investigation of what we would now recognize as ultramafic sequences was in its infancy and it is one of Bonney's most significant contributions that he recognized that the serpentine was an alteration product, not a primary lithologic mineralogy. This opened the door to future work. The status of understanding of metamorphic petrology at this era is the subject of a manuscript currently in review in Earth Sciences History entitled Late 19[th] century understandings of the origins of metamorphic rocks and their classification. For the review the abstract of that paper is as follows.

During the last decades of the 19[th] century scientific understandings of the processes and products of metamorphism were both incomplete and highly fragmented by the diversity of petrologic views held by a group of active and highly knowledgeable workers. While a generally agreed upon definition of metamorphism had by that time begun to coalesce, a consensus on the structure and organization, indeed even on the essential characteristics, of the classification of metamorphic rocks had not yet emerged. Competing schemes were developed that considered chemico-mineralogical, textural and mineralogical, and process-based understanding as the most significant criteria of classification. Despite the absence of a single classification system, two of what were widely recognized as among the most significant processes of metamorphic alteration received intense consideration during this interval. The first included those mineralogical and chemical changes driven by the thermal alteration of rocks found in close proximity to igneous intrusions and was termed thermal or contact metamorphism. The second was characterized by mineralogical and textural changes induced by deviatoric forces resulting in brittle and ductile deformation of protoliths and was known as dynamic metamorphism. Those two domains of metamorphic petrology were the subject of active research within a fertile and dynamic intellectual setting that was augmented by rapid advances in geochemical and optical techniques. Further, the exploration of those processes drew some of the most decorated and impassioned petrologists of Britain to debate and define the future of metamorphic petrology.

Expanding the current paper on the Lizard to include a representative summary of metamorphic petrology would significantly lengthen the paper. The topic editor has asked that I remove the references to the Drummond 2026 manuscript and I have done so.

The reviewer also asks "How did people interpret thrust tectonics more broadly?" I have tried to make it more clear in the text that there was almost no understanding of the details of the deformation of the Lizard at this time. The presence of faulting was recognized but beyond a few dip direction measurements, none of the works considered provided any information on the sense of motion on those fault boundaries or the magnitudes of displacement. Bonney concluded that the displacements were minimal. Since there was no interpretation of the structural relationships present in the Lizard, there is really nothing more to say about it in this paper. Processes of serpentinization is the key topic of Bonney, and that is why it is covered in great detail.

The reviewer breaks out specific comments by section of the paper:

Abstract – I have made these changes and given more background on Bonney in the abstract

Introduction – I have not added a second map, but I have included latitude and longitude in Fig. 1. I hope this is satisfactory. I have made the arrows smaller on the tectonostratigraphic section.

I have capitalized all occurrences of Lizard District.

2.1 Majendie

I have added to additional locations to Figure 1, Loe-Bar on the northwest and Helford estuary on the north east.

I have added a statement that makes clear the congruency is in the location of major lithologic boundaries.

2.3 Rogers and 2.4 De la Breche
I have acknowledged these details.

2.5 King and Rowney

Spelling corrected
Image included as Fig. 3

3.1 Survey of the west and east coasts

Spelling corrected
The modern interpretation is provided in Mackay- Champion et al 2024. I have not reproduced their conclusions here.
Spelling corrected
The trap dyke is an odd term used by Bonney and others to describe the basaltic dykes of the northeast coast, I have clarified this in the figure captions and text

3.3 Temporal relationships

I have highlighted the advances of Bonney's approach
See above

3.4 Microscopic examinations

Spelling corrected
Language corrected
Corrected

4.1 Field and microscopic survey of the schist

I agree, this is really astonishing and I cannot give a very good explanation for this error. One thought that comes to mind is that in the Lands End and Lake District examples of metamorphism of volcanic rocks, the workers were able to walk out changes from highly altered near the intrusion to largely unaltered at a distance from the junction. Bonney could not do this in the Lizard but may have been overly interpreted the textures he observed using the thermal metamorphism model that had been advanced elsewhere. I have not substantially addressed this misinterpretation further.

In summary, I have found the comments and suggestions of the reviewers very helpful and I have incorporated their suggested changes in all cases other than the few minor items described above.

---

## Author Response (AR2)

Final Author's Response 11/24/25.

I have made two very small edits to the manuscript.

First, I have corrected the postal code number.

Second, I have moved the reviewer acknowledgements in accordance with house style.

Thank you so much for the great reviews and the opportunity to share my work with your readership!

All the best

Carl

Carl N Drummond